# Inflammation-induced Id2 promotes plasticity in regulatory T cells

Sung-Min Hwang[1,2], Garima Sharma[1,2], Ravi Verma[1], Seohyun Byun [1,2], Dipayan Rudra[1,2] & Sin-Hyeog Im [1,2,3]

$T_H17$ cells originating from regulatory T ($T_{reg}$) cells upon loss of the $T_{reg}$-specific transcription factor Foxp3 accumulate in sites of inflammation and aggravate autoimmune diseases. Whether an active mechanism drives the generation of these pathogenic 'ex-Foxp3 $T_H17$' cells, remains unclear. Here we show that pro-inflammatory cytokines enhance the expression of transcription regulator Id2, which mediates cellular plasticity of $T_{reg}$ into ex-Foxp3 $T_H17$ cells. Expression of Id2 in in vitro differentiated $iT_{reg}$ cells reduces the expression of *Foxp3* by sequestration of the transcription activator E2A, leading to the induction of $T_H17$-related cytokines. $T_{reg}$-specific ectopic expression of Id2 in mice significantly reduces the $T_{reg}$ compartment and causes immune dysregulation. Cellular fate-mapping experiments reveal enhanced $T_{reg}$ plasticity compared to wild-type, resulting in exacerbated experimental autoimmune encephalomyelitis pathogenesis or enhanced anti-tumor immunity. Our findings suggest that controlling Id2 expression may provide a novel approach for effective $T_{reg}$ cell immunotherapies for both autoimmunity and cancer.

[1] Academy of Immunology and Microbiology (AIM), Institute for Basic Science (IBS), Pohang 37673, Republic of Korea. [2] Division of Integrative Biosciences and Biotechnology, Pohang University of Science and Technology (POSTECH), Pohang 37673, Republic of Korea. [3] Department of Life Sciences, Pohang University of Science and Technology (POSTECH), Pohang 37673, Republic of Korea. Correspondence and requests for materials should be addressed to D.R. (email: rudrad@ibs.re.kr) or to I.S.H. (email: iimsh@postech.ac.kr)

Regulatory T ($T_{reg}$) cells are a unique population of CD4+ T-cells essential for maintaining immune homeostasis[1–4]. Stable expression of the X-chromosome encoded transcription factor Foxp3 distinguishes $T_{reg}$ cells from other T-cell lineages[5,6], and is a prerequisite for maintaining their suppressive properties. Functional deficiencies in Foxp3 results in overt lymphoproliferation and systemic autoimmune features both in mice and human patients characterized by the "scurfy" phenotype and immunodysregulation polyendocrinopathy enteropathy X-linked (IPEX) syndrome respectively[7,8].

Classically, each CD4+ T helper ($T_H$) subsets are viewed as terminally differentiated and specialized for their discriminative functions. However, it has been suggested that plasticity within effector CD4+ $T_H$ cell populations is capable of exerting flexible immune responses under various physiological conditions[9,10]. Several reports have revealed that under inflammatory and autoimmune disease conditions, loss of Foxp3 results in high degree conversion of $T_{reg}$ cells towards a $T_H$17-like "ex-Foxp3 $T_H$17" phenotype[11–15]. Consequently, converted ex-Foxp3 $T_H$17 cells become more pathogenic and contribute to the progression and severity of the disease. The molecular basis of this plasticity remains to be fully characterized.

Id proteins (Id1-Id4) are inhibitors of helix-loop-helix (HLH) DNA binding transcription factors and play diverse roles in immune cell development and function. Id proteins are known to mainly inhibit DNA-binding activities of E proteins, a prevalent HLH domain containing family of transcription factors that include E2A, E2-2, and HEB. Id proteins, which lack any detectable DNA-binding domain, act by interfering with the formation of active homo- or hetero-dimers by E-proteins, a prerequisite for their DNA binding and transcription related activities[16–18].

Together with Id3, Id2 has been shown to be an important regulator controlling multiple aspects of CD4+ T cell differentiation. Recently published data suggest that Id2 enhances $T_H$1, but attenuates $T_{FH}$ cell differentiation[19]. Simultaneous deletion of Id2 and Id3 results in defect in maintenance and localization, and enhanced differentiation towards T follicular regulatory ($T_{FR}$) subtype of $T_{reg}$ cells[20]. Furthermore, mice with T cell specific deletion of Id2 display resistance towards experimental autoimmune encephalomyelitis (EAE)[21], raising the possibility of its potential function in $T_H$17 mediated pathogenesis.

Here we show that Id2 is induced in $T_{reg}$ cells under various inflammatory settings. Ectopic expression of Id2 results in reduced expression of Foxp3 and enhanced $T_H$17 cell-related cytokines in in vitro induced $T_{reg}$ (i$T_{reg}$) cells. In mice, $T_{reg}$ cell-specific overexpression of Id2 causes $T_{reg}$ instability, and induces susceptibility to EAE pathogenesis and spontaneous age-related autoimmunity. IL-1β and IL-6 signaling mediated STAT3/IRF4/BATF transcriptional activity is found to be responsible for Id2 induction, which in turn inhibits the binding of E2A to the *Foxp3* locus, thereby influencing $T_{reg}$ stability. In a melanoma model of cancer, temporal overexpression of Id2 in $T_{reg}$ cells suppresses tumor growth in mice. Our data thus identify a novel cell intrinsic molecular mechanism underlying $T_{reg}$ cell plasticity with potential therapeutic significance in both autoimmunity and cancer.

## Results

**Enhanced Id2 expression in ex-Foxp3 $T_H$17 cells**. As an initial approach to identify critical factor(s) that might affect the plasticity of $T_{reg}$ cells, we re-analyzed previously published microarray data and compared gene-expression profiles of $T_{reg}$ and ex-Foxp3 $T_H$17 cells[14]. Since $T_H$17 cells and ex-Foxp3 $T_H$17 cells have similar phenotypes, albeit being of different origin, we also used the $T_H$17 cell gene expression profile alongside for this analysis. We focused on a set of 449 genes which, while are expressed at a low level in i$T_{reg}$ cells in comparison to $T_H$0, are de-repressed in $T_H$17 as well as ex-Foxp3 $T_H$17 cells (Supplementary Fig. 1a). Among these, by employing Gene Ontology (GO) analysis, we focused on the genes that are related to immune system and /or are involved in regulation of transcription (Supplementary Fig. 1b). We identified Id2 as a putative target which is most prominently expressed both in ex-Foxp3 $T_H$17 cells and $T_H$17 cells compared to i$T_{reg}$ cells (Supplementary Fig. 1c). Gene expression data from two other independent studies also implicated Id2 as a potential target. First, Id2 expression in ex-Foxp3 $T_H$17 was dramatically increased compared to $T_{reg}$ cells under inflammatory conditions caused by asthma[15] (Supplementary Fig. 1d). Second, Id2 expression negatively correlated with enhanced stability of Rorγt+Foxp3+ $T_{reg}$ cell lineage (Supplementary Fig. 1e)[22].

By employing in vitro culture conditions, we next wanted to determine whether enhanced Id2 expression indeed correlates with the $T_H$17 or ex-Foxp3 $T_H$17 differentiation process. We sorted CD4+CD25−CD62L$^{hi}$CD44$^{lo}$ naive T cells and differentiated them into $T_H$17, i$T_{reg}$, or ex-Foxp3 $T_H$17 subsets (Fig. 1a). Indeed, Id2 mRNA was found to be highly induced in $T_H$17 cells and ex-Foxp3 $T_H$17 cells compared to i$T_{reg}$ cells (Fig. 1b). A time course analysis to determine Id2 protein expression under similar experimental conditions revealed that its expression in $T_H$17 cells commences at 48 h, a time point at which Rorγt expression starts declining, and $T_H$17 specific cytokines IL-17A and IL-17F are expressed most (Fig. 1c). Id2 protein expression in i$T_{reg}$ differentiation condition however was almost negligible, and the low level of the protein expressed at earlier time points was further reduced when detectable expression of Foxp3 and IL-10 was observed (Fig. 1d). More interestingly, under ex-Foxp3 $T_H$17 polarizing condition, Id2 protein expression was found to gradually increase, mirroring a concomitant reduction in Foxp3 protein level with time (Fig. 1e, upper panel). Furthermore, under these conditions the cells displayed decrease in IL-10 expression and an increase in IL-17A expression, suggesting a bonafide conversion to $T_H$17 phenotype (Fig. 1e, lower panel).

In order to rule out the possibility that the absence of a suitable marker for $T_{reg}$ cells in the above set of experiments may have influenced our interpretation of the results, we also repeated these experiments by using CD4+CD62L$^{hi}$CD44$^{lo}$Foxp3$^{Thy1.1−}$ T cells sorted from Foxp3$^{Thy1.1}$ reporter mice, in which Thy1.1 allele is knocked into *Foxp3* locus[23], as the starting naive population (Supplementary Fig. 2a). Intracellular staining for relevant transcription factors and $T_H$17 related cytokines essentially led to identical conclusions (Supplementary Fig. 2b, c). Furthermore, including an additional culture condition in which TGF-β was excluded from ex-Foxp3 $T_H$17 skewing cocktail revealed that while TGF-β is dispensable for Id2 induction and Foxp3 downregulation, it is required for optimum induction of IL-17A and subsequent downregulation of IL-10 (Supplementary Fig. 2c). Rorγt on the other hand appeared to be only minimally induced only in the presence of TGF-β (Supplementary Fig. 2b).

**Id2 promotes i$T_{reg}$ to ex-Foxp3 $T_H$17 differentiation in vitro**. Based on the above results, we hypothesized that Id2 is likely to negatively affect the differentiation process of i$T_{reg}$ cells, and examined the effect of Id2 overexpression during $T_H$17 and i$T_{reg}$ differentiation conditions. Control vector, or a vector encoding Id2 cDNA were retrovirally transduced in T cells, which were skewed towards $T_H$17 or i$T_{reg}$ conditions (Supplementary Fig. 3a). Retroviral expression of Id2 mRNA and protein was confirmed in transduced cells by RT-qPCR and western blot analysis respectively (Fig. 2a). Provision of Id2 in $T_H$17 skewing

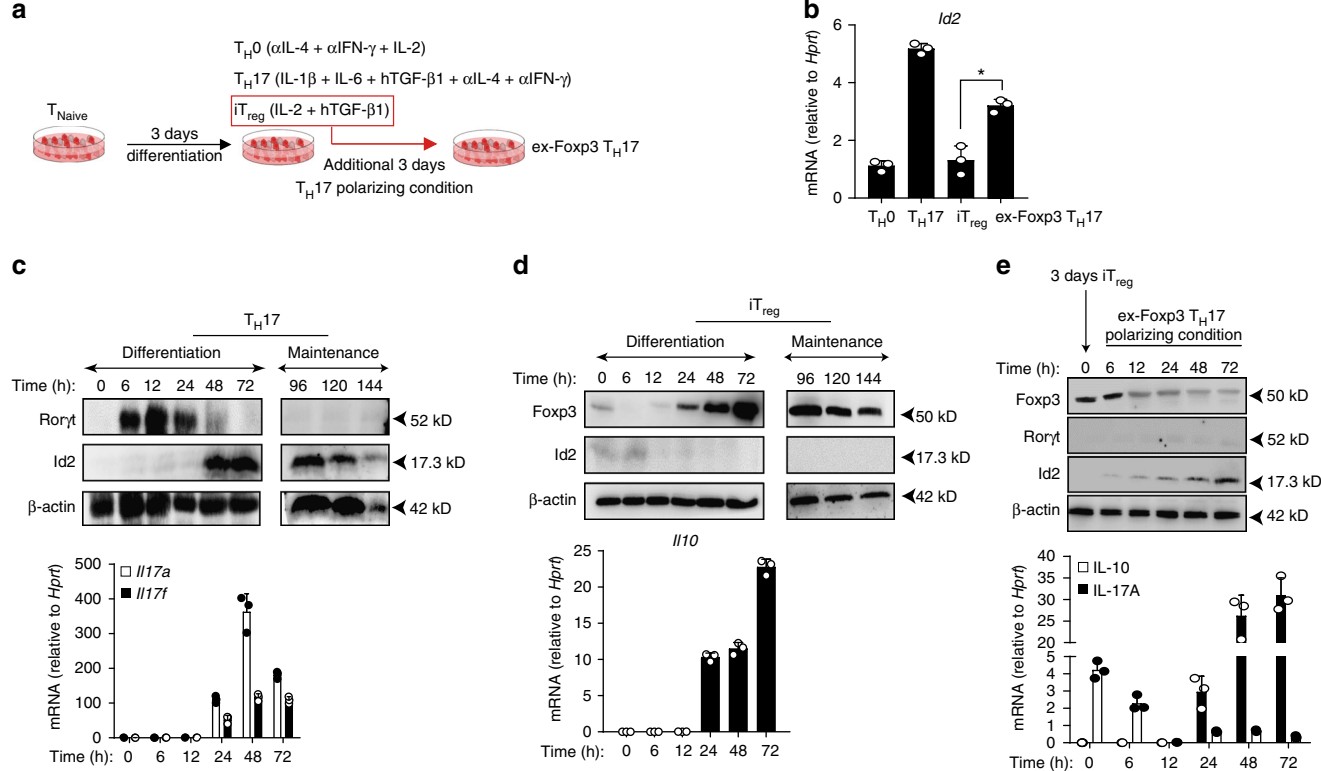

**Fig. 1** Id2 expression is enhanced during a conversion to $T_H17$ phenotype from $iT_{reg}$ cells. **a** In vitro polarization scheme of $T_H0$, $T_H17$, $iT_{reg}$ and ex-Foxp3 $T_H17$ cells. **b** RT-qPCR analysis of Id2 mRNA in in vitro generated $T_H0$, $T_H17$, $iT_{reg}$ and ex-Foxp3 $T_H17$ cells; results are presented relative to Hprt expression ($n = 3$) **c, d** Western blot analysis to determine the kinetics of indicated proteins during $T_H17$ (**c**, top panel) and $iT_{reg}$ (**d**, top panel) cell differentiation and maintenance conditions. β-actin is used as loading control. RT-qPCR analysis of Il17a, Il17f (**c**, lower panel) and Il10 (**d**, lower panel) mRNA during each $T_H17$ and $iT_{reg}$ cell differentiation conditions; results are presented relative to Hprt expression ($n = 3$). **e** Sorted $CD4^+$ naive T cells were activated in vitro under $iT_{reg}$ cell differentiation condition for 3 days. After 3 days, $iT_{reg}$ cells were re-stimulated in vitro in ex-Foxp3 $T_H17$ polarizing conditions for additional 3 days. Western blot analyses for the indicated proteins were performed after harvesting the cells at indicated time points (top panel). RT-qPCR analysis of Il10 and Il17a mRNA under similar experimental conditions is shown in the lower panel; results are presented relative to Hprt expression ($n = 3$). *$P < 0.05$ (Student's t-test). Data are representative two independent experiments (error bars, s.d.)

condition was found to result in enhanced maintenance of Rorγt mRNA and protein (Supplementary Fig. 3b, c). IL-17F and IL-22 expressions were also enhanced at mRNA and protein level. Expression of IL-17A, while increased at mRNA level, was found to be comparable in terms of protein expression between vector transduced and Id2 transduced groups at this point, presumably reflecting delayed translation (Supplementary Fig. 3d, e). Conversely, under $iT_{reg}$ differentiation condition, Id2 expression resulted in dramatic reduction in the expression of Foxp3 mRNA and protein, compared to vector control (Fig. 2b, c). Furthermore, $iT_{reg}$ cells, in the presence of excess Id2 displayed enhanced expression of $T_H1$ and $T_H17$ related cytokines IFN-γ, IL-17A, IL-17F, and IL-22 (Fig. 2d, e). Expression of mRNAs encoding $T_H2$ related cytokines IL-4, IL-5, and IL-13 as well as that of IL-10, TGF-β, and TNF remained unaltered.

In the above experimental setup T naive cells were differentiated under $T_H17$ or $T_{reg}$ conditions prior to retroviral overexpression of Id2. In order to rule out any discrepancies arising from such non-homogeneous culture conditions, we also performed this experiment under modified conditions whereby cells were activated in the absence of any cytokines one day prior to retroviral transductions. Differentiation was ensued after retroviral transduction. This differentiation condition appeared to be suboptimal, particularly for the cells harboring the retrovirus ($GFP^+$ cells), and resulted in substantially compromised differentiation of the transduced precursors towards respective effector cell types. Nevertheless, overall conclusions

from these experiments remained the same remained the same (Supplementary Fig. 4a–g). Taken together, these results indicated that enhanced Id2 expression leads to unstable $iT_{reg}$ lineage commitment, and convert them into $T_H17$-like cells.

**$T_{reg}$-specific ectopic expression of Id2 in vivo.** To define the functional role of Id2 in regulation of plasticity of $T_{reg}$ cells, we generated $P_{CMV}$-lsl-$Id2^{EmGFP}$ mice ($Id2^{EmGFP}$ in short) (Fig. 3a) in which CMV promoter driven cDNA encoding Id2 fused with Emerald Green Fluorescent Protein (EmGFP) is preceded by loxP flanked mCherry. We crossed $Id2^{EmGFP}$ mice with $Foxp3^{YFP-Cre}$ mice[24] and generated $Id2^{EmGFP}Foxp3^{YFP-Cre}$ mice (Fig. 3a) in which deletion of the loxP sites would remove the stop codon of mCherry cDNA, resulting in the expression of Id2-EmGFP fusion protein specifically in $T_{reg}$ cells (Fig. 3a, b). Notably, the transgene locus in these mice is also equipped with a Tetracycline Operator (TetO) sequence, providing it with the provision of being temporally regulated in a Doxycycline (Dox) dependent manner upon co-expression of the Tetracycline Repressor protein (TetR) (Fig. 3a and see later). As a proof of principle, compared to control $Foxp3^{YFP-Cre}$, in the $Id2^{EmGFP}Foxp3^{YFP-Cre}$ mice we observed a shift in the $CD4^+CD25^+$ $T_{reg}$ cells towards a $GFP^{hi}$ population. Expression of Id2 was confirmed in this population by intracellular staining (Fig. 3c).

$Id2^{EmGFP}Foxp3^{YFP-Cre}$ mice were born at the expected mendelian ratios and showed no clinical signs of sickness till

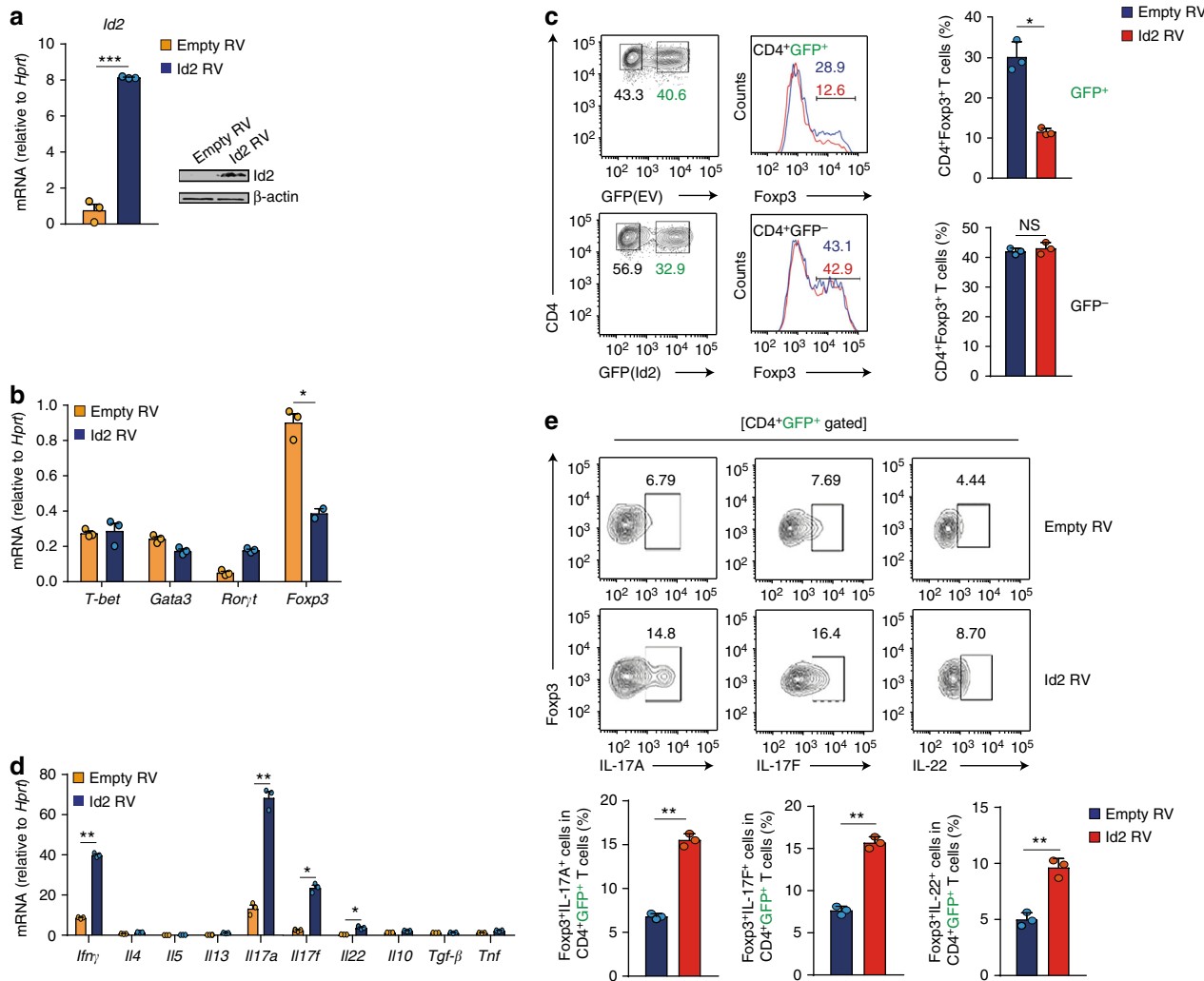

**Fig. 2** Overexpression of Id2 in vitro fails to stabilize iT$_{reg}$ lineage commitment, and convert them into T$_H$17-like cells. **a** Naive CD4$^+$ T cells were sorted from wild-type C57BL/6 (B6) mice, and transduced with control vector (Empty RV) or vector encoding Id2 cDNA (Id2 RV) under iT$_{reg}$ differentiation condition. After 3 days cells were harvested and Id2 expression was measured by RT-qPCR ($n = 3$, per group) and Western blot analysis. **b** Comparison of mRNA expression for *T-bet*, *Gata3*, *Rorγt* and *Foxp3* between Empty RV and Id2 RV transduced T cells after 3 days post spinfection under iT$_{reg}$ differentiation condition ($n = 3$, per group). **c** Flow cytometry analysis of Foxp3 expression between Empty RV or Id2 RV transduced (GFP$^+$) and non-transduced (GFP$^-$) iT$_{reg}$ cells ($n = 3$, per group). **d** Comparison of mRNA expression for genes encoding the indicated cytokines between Empty RV and Id2 RV transduced T cells after 3 days post spinfection under iT$_{reg}$ differentiation condition ($n = 3$, per group). **e** Flow cytometry analysis of IL-17A, IL-17F and IL-22 from Empty RV or Id2 RV transduced CD4$^+$GFP$^+$ iT$_{reg}$ cells ($n = 3$, per group). NS, not significant, *$P < 0.05$, **$P < 0.005$ (Student's $t$-test). All data are representative two or three independent experiments (error bars, s.d.)

6–8 weeks of age (Supplementary Fig. 5a). Analysis of these mice at this age revealed only moderate increase in frequencies and activation state as determined by CD62L$^{lo}$CD44$^{hi}$ CD4$^+$ and CD8$^+$ T cell compartments (Supplementary Fig. 5b–d). Furthermore, there was only a marginal decrease in CD4$^+$Foxp3$^+$ T$_{reg}$ frequency particularly in the peripheral lymphoid organs (Supplementary Fig. 5e), suggesting that at least under steady state condition, T$_{reg}$-specific ectopic expression of Id2 does not lead to enhanced loss of Foxp3 at an early age.

To determine whether in contrast to steady state, exposing young *Id2*$^{EmGFP}$*Foxp3*$^{YFP-Cre}$ mice to inflammatory conditions would lead to enhanced T$_{reg}$ cell plasticity, we next employed a model of experimental autoimmune encephalomyelitis (EAE), an animal model for multiple sclerosis in which T$_{reg}$ cell instability and conversion to effector T cells is well documented[13,25,26]. For this we crossed *Id2*$^{EmGFP}$*Foxp3*$^{YFP-Cre}$ mice with mice harboring R26$^{lsl-tdTomato}$ (R26T in short) locus (resultant mice called *Id2*$^{EmGFP}$R26T*Foxp3*$^{YFP-Cre}$ or control R26T*Foxp3*$^{YFP-Cre}$), in

which the cDNA encoding tdTomato is preceded by stop codon flanked by loxP sites, and is expressed upon Cre-mediated recombination. The fate of Foxp3 expression in tdTomato$^+$ cells in these mice could be determined by monitoring YFP-Cre expression. Under this experimental condition upon induction of EAE, as expected even in control R26T*Foxp3*$^{YFP-Cre}$ mice, increased Id2 expression was observed in the tdTomato$^+$YFP$^-$ (ex-T$_{reg}$) cells, compared to tdTomato$^+$YFP$^+$ (T$_{reg}$) cells, confirming a correlation between increased Id2 expression and downregulation of Foxp3 (Fig. 3d, e). Importantly, the extent of Id2 expression in these ex-T$_{reg}$ cells was comparable to that observed in T$_{reg}$ cells derived from *Id2*$^{EmGFP}$*Foxp3*$^{YFP-Cre}$ mice, suggesting that the ectopic T$_{reg}$ specific expression of Id2 in these mice remains within physiological limit (compare Id2 expression in Fig. 3c, e).

Indeed, compared to controls, the *Id2*$^{EmGFP}$R26T*Foxp3*$^{YFP-Cre}$ mice displayed significantly enhanced disease score throughout the course of the experiment (Fig. 3f, g). This was well correlated

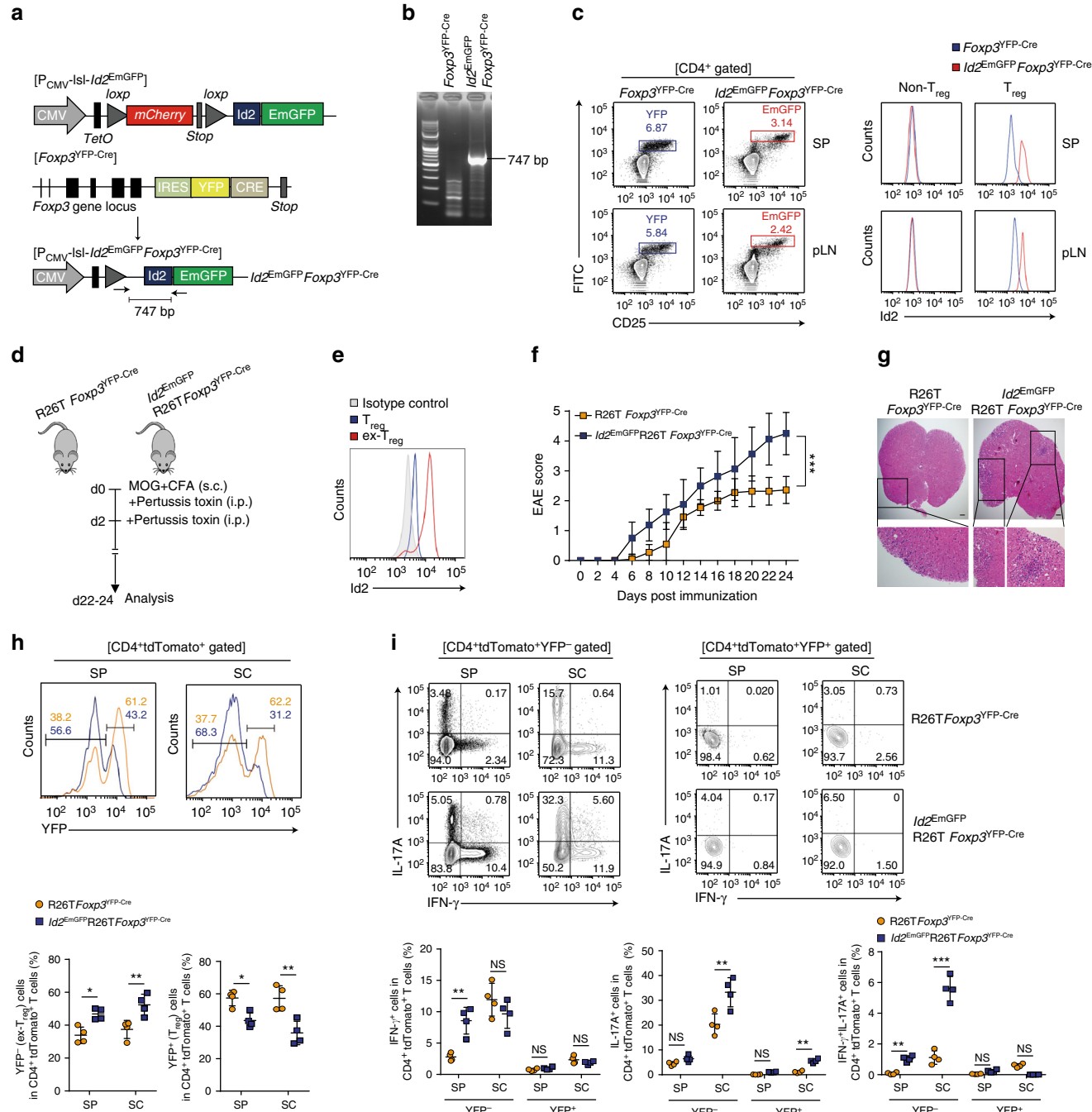

**Fig. 3** Mice with $T_{reg}$-specific ectopic expression of Id2 displayed enhanced conversion to ex-Foxp3 $T_H17$ cells from $T_{reg}$ cells after induction of EAE. **a** Schematic representation the mouse model employed. Converging arrows in the lower panel indicate genotyping primers. **b** Genotyping PCR to detect the presence of Id2-EmGFP transgene in $Id2^{EmGFP}Foxp3^{YFP-Cre}$ mice. **c** Flow cytometry analysis of YFP and EmGFP expression in $CD4^+CD25^+$ T cells from spleen (SP) and peripheral lymph nodes (pLN) of $Foxp3^{YFP-Cre}$ and $Id2^{EmGFP}Foxp3^{YFP-Cre}$ mice (left panel). Id2 expression was assessed between non-$T_{reg}$ and $T_{reg}$ cells by intracellular staining (right panel). **d** Schematic of experimental EAE model. **e** Representative flow cytometry analysis of Id2 expression between tdTomato$^+$YFP$^+$ ($T_{reg}$) and tdTomato$^+$YFP$^-$(ex-$T_{reg}$) cells in CD4$^+$ T cells from 6 to 8 week-old R26T$Foxp3^{YFP-Cre}$ mice on day 24 after induction of EAE. **f** Mean clinical scores in mice after induction of EAE (R26T$Foxp3^{YFP-Cre}$; $n = 12$, $Id2^{EmGFP}$R26T$Foxp3^{YFP-Cre}$; $n = 9$). **g** Representative Spinal Cord (SC) sections and hematoxylin and eosin (H&E) staining from EAE induced mice at day 24 after immunization (scale bar, 100 μm). **h** Flow cytometry analysis and percentages of Foxp3 sufficient (YFP$^+$) and deficient (YFP$^-$) populations among CD4$^+$tdTomato$^+$ in SP and SC from R26T$Foxp3^{YFP-Cre}$ and $Id2^{EmGFP}$R26T$Foxp3^{YFP-Cre}$ mice at day 22–24 after induction of EAE. **i** Representative flow cytometry analysis and percentages of IFN-γ$^+$, IL-17A$^+$ and IFN-γ$^+$IL-17A$^+$ in CD4$^+$tdTomato$^+$ YFP$^-$ or CD4$^+$tdTomato$^+$ YFP$^+$ T cells in total SP and SC cells stimulated with phorbol myristate acetate (PMA) and ionomycin for 6 h. NS, not significant, *$P < 0.05$, **$P < 0.005$, ***$P < 0.001$ (Student's $t$-test). All data are representative of two independent experiments (error bars, s.d.)

with heightened instability of Foxp3 in Id2 expressing cells specifically in spinal cord of diseased mice, as determined by enhanced loss of YFP-Cre expression within tdTomato$^+$ cells in $Id2^{EmGFP}$R26T$Foxp3^{YFP-Cre}$ mice compared to controls (Fig. 3h). Furthermore, Id2 expressing tdTomato$^+$YFP$^-$ cells displayed heightened production of IL-17A, as well as were simultaneous producers of IL-17A and IFN-γ (Fig. 3i).

These results strongly indicated that ectopic expression of Id2 in $T_{reg}$ cells indeed results in heightened instability of Foxp3 expression. However, provided that under certain activated conditions naive T cells are known to express Foxp3, albeit in a transient manner[27], there remained a possibility that the increased frequency of tdTomato$^+$YFP$^-$ cells upon Id2 over-expression is a result of enhanced accumulation of such promiscuous Foxp3 expressing naive T cells. In order to clarify this issue, we sorted CD4$^+$FITC$^-$tdTomato$^-$CD44$^{lo}$CD62L$^{hi}$ naive T cells from $Id2^{EmGFP}$R26T$Foxp3^{YFP-Cre}$ or R26T$Foxp3^{YFP-Cre}$ mice. In order to determine the extent of promiscuous Foxp3 expression from these cells, they were cultured in the presence of TGF-β blocking antibody, a condition that was shown to induce transient Foxp3 induction in vitro[27]. The extent of promiscuous Foxp3 expression was found to be identical in T cells derived from both mice. A control iT$_{reg}$ induction assay in the presence of TGF-β, as expected, yielded more iT$_{reg}$ cells from T naive cells derived from R26T$Foxp3^{YFP-Cre}$ (Supplementary Fig. 6a). Furthermore, when on the other hand, CD4$^+$FITC$^+$tdTomato$^+$ T$_{reg}$ cells were sorted from these mice and cultured in vitro, the T$_{reg}$ cells derived from $Id2^{EmGFP}$R26T$Foxp3^{YFP-Cre}$ displayed enhanced instability compared to those derived from R26T$Foxp3^{YFP-Cre}$ mice (Supplementary Fig. 6b). Taken together these results strongly indicated that Treg specific ectopic expression of Id2 results in loss of Foxp3 expression, rather than increased stability of transient Foxp3$^+$ naive T cells.

While the $Id2^{EmGFP}Foxp3^{YFP-Cre}$ mice showed little immune dysregulation at a young age, it could be possible that with time due to prolonged presence of Id2, T$_{reg}$ cells in these mice loose Foxp3, resulting in visible autoimmune phenotype. In concert to this hypothesis, indeed at a relatively older age of 12–16 weeks, these mice developed overt autoimmunity, characterized by massive splenomegaly and lymphadenopathy (Fig. 4a). Histological analysis showed dramatic infiltration of lymphocytes into multiple organs such as skin, lung and liver (Fig. 4b). Notably, they had increased percentages of CD4$^+$ and CD8$^+$ T cells that were associated with dramatically increased frequencies of CD62L$^{lo}$CD44$^{hi}$ effector memory populations (Fig. 4c–e). Importantly, while CD4$^+$Foxp3$^+$ T$_{reg}$ frequency in the thymus remained largely intact, T$_{reg}$ cell frequency was found to be critically reduced in the pLN and spleen of $Id2^{EmGFP}Foxp3^{YFP-Cre}$ mice (Fig. 4f).

Together, these data strongly suggest an active role of Id2 in T$_{reg}$ plasticity and pathogenesis accompanied with inflammation caused by autoimmune conditions such as EAE. Furthermore, while mice with T$_{reg}$-specific overexpression of Id2 remains disease free at an early age, steady loss of Foxp3 expression eventually results in severely diminished T$_{reg}$ compartment and results in systemic autoimmunity later in their life.

**IL-1β and IL-6 mediated STAT3/IRF4/BATF induce Id2.** Next we sought to determine the cellular and molecular events responsible for driving Id2 expression in T$_{reg}$ cells under inflammatory conditions. In recent years, the roles of two major cytokines, IL-1β and IL-6, in driving T$_{reg}$ plasticity resulting in its phenotypic conversion to T$_H$17-like cells is well documented[28–30]. We therefore hypothesized that IL-1β and IL-6 signaling may induce Id2 expression in iT$_{reg}$ cells. Indeed, in agreement with

this hypothesis we observed that the receptors for both these cytokines, IL-1r1 and IL-6rα were dramatically upregulated under in vitro culture conditions specifically in ex-Foxp3 T$_H$17 cells derived from $Foxp3^{Thy1.1}$ mice (in which the Thy1.1 allele is knocked into $Foxp3$ locus[23]) (Fig. 5a). Conversely, when in vitro differentiated iT$_{reg}$ cells were treated with IL-1β and IL-6, either by themselves in a dose dependent manner (Supplementary Fig. 7a), or in combination, resulted in significant upregulation of $Id2$ transcript and protein expression (Fig. 5b, left panel and 5c). As expected, this was associated with a concomitant reduction in Foxp3 mRNA and protein, and enhanced IL-17A expression (Fig. 5b right panel and Supplementary Fig. 7b). These results strongly suggested that signaling events downstream of IL-1β and IL-6 are major contributors to enhanced Id2 expression in ex-Foxp3 T$_H$17 cells.

In order to determine the transcription factors responsible for Id2 upregulation, we next re-analyzed published ChIP-seq data and asked whether any of the five transcription factors previously implicated as "core" determinants of T$_H$17 gene expression program[31], is physically associated with $Id2$ gene locus. Indeed, among these, STAT3, IRF4 and BATF were found to be strongly associated to Id2 locus specifically in T$_H$17 cells, whereas the bindings for the other two factors Rorγt and Maf were relatively less prominent (Supplementary Fig. 7c). Further substantiating this observation, we detected potential binding sites for these transcription factors throughout the upstream gene regulatory regions of $Id2$ transcription start site (TSS) (Fig. 5d). Finally, by employing luciferase assay based transcription reporter system, we identified a specific region −582 to −147 base pairs upstream of $Id2$ TSS to be the key region responsive to these transcription factors, most effectively in cooperation with each other (Supplementary Fig. 7d and Fig. 5e, f). Together, these results indicated that IL-1β and IL-6 signaling-mediated STAT3, IRF4, and BATF can induce Id2 expression in T$_{reg}$ cells.

**Id2 sequesters E2A from binding to $Foxp3$ promoter.** We next assessed how enhanced level of Id2 in the presence of inflammatory cytokines might repress $Foxp3$ expression. Id proteins, upon forming non-functional heterodimers, are well known naturally occurring dominant-negative inhibitors of E-protein transcription activators (e.g., E2A, HEB, and E2-2)[16–18]. Since E2A, one of the best characterized members of the E-protein family, has been previously implicated in promoting transcription of $Foxp3$ by being physically associating with multiple regions of its promoter[32], we hypothesized that Id2 de-repression would potentially sequester E2A, thereby repressing Foxp3 expression. In agreement with this possibility, unlike $Foxp3$ and $Id2$, the transcript level of the gene encoding E2A was found to be only minimally affected in iT$_{reg}$ compared to ex-Foxp3 T$_H$17 cells (Fig. 6a), suggesting a post-transcriptional alteration of E2A's activity might be instrumental.

Analysis of the DNA sequences encompassing −1702 to +174 of Foxp3 TSS revealed three strong E-protein binding motifs, 1593−1584, 1295–1286, and 837–829 bps upstream to TSS. (Supplementary Fig. 8a). ChIP-qPCR analysis confirmed E2A binding to all of these sites compared to a region with no predicted E-protein binding motif in iT$_{reg}$ cells (Fig. 6b). More importantly, the relative occupancy of E2A to all these sites were significantly reduced in ex-Foxp3 T$_H$17 cells, suggesting functional sequestration of E2A in the presence of Id2 (Supplementary Fig. 8a and Fig. 6b). In agreement to this possibility, in a luciferase based assay system in which the luciferase reporter gene was driven by −1702 to +174 bps encompassing $Foxp3$ promoter, provision of E2A increased luciferase activity, which was significantly repressed upon co-transfection with Id2 (Fig. 6c).

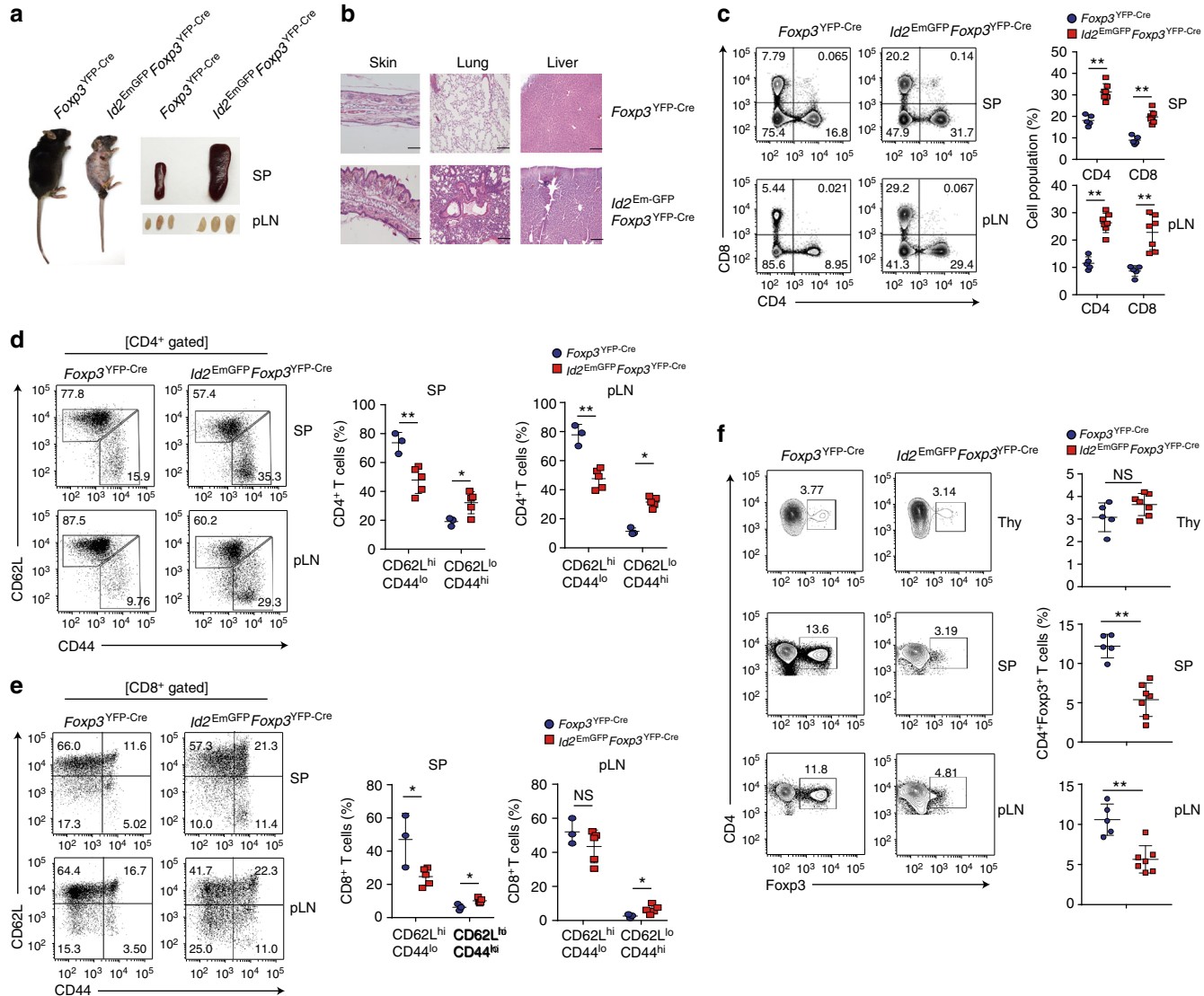

**Fig. 4** $T_{reg}$-cell-specific overexpression of Id2 results in systemic autoimmunity in older mice. **a** A representative picture of 12–16 week-old $Foxp3^{YFP-Cre}$ and $Id2^{EmGFP}Foxp3^{YFP-Cre}$ mice (*left*) and spleen (SP) and peripheral lymph nodes (pLN) derived from them (*right*). **b** Representative hematoxylin and eosin (H&E)-stained skin, lung, liver sections from 12–16 weeks old $Foxp3^{YFP-Cre}$ and $Id2^{EmGFP}Foxp3^{YFP-Cre}$ mice (scale bar, 100 μm). **c** Representative FACS plots and percentages of CD4+ and CD8+ T cells from SP and pLN of 12–16 weeks old $Foxp3^{YFP-Cre}$ and $Id2^{EmGFP}Foxp3^{YFP-Cre}$ mice. **d, e** Representative FACS plots and percentages of CD62L$^{hi}$CD44$^{lo}$ (naive) and CD62L$^{lo}$CD44$^{hi}$ (effector/memory) cells among CD4+ and CD8+ T cells in SP and pLN 12–16 week-old mice. **f** Representative FACS plots and percentages of Foxp3+ $T_{reg}$ cells in thymus, SP and pLN from 12–16 week-old mice. NS, not significant, *$P < 0.05$, **$P < 0.005$ (Student's *t*-test). All data are representative three independent experiments (error bars, s.d.)

Furthermore, mutating the E-box binding motifs rendered the reporter unresponsive to both E2A and Id2, confirming specificity for E2A (Supplementary Fig. 8b and Fig. 6d). Taken together, these results strongly indicated that increased Id2 can inhibit Foxp3 expression through modulating E2A accessibility to *Foxp3* promoter.

**Id2 dependent $T_{reg}$ plasticity enhances antitumor immunity.** $T_{reg}$ cells are known to promote tumor progression by creating an immunosuppressive tumor environment[33–35]. Increased $T_{reg}$ cell numbers and function has therefore been associated with suppression of anti-tumor immunity[36,37]. Also, a low $T_{reg}$: $T_{eff}$ ratio is associated with better prognosis in cancer[38–42].

Since we observed increased inflammation and age dependent plasticity of $T_{reg}$ cell populations in peripheral compartment of $Id2^{EmGFP}Foxp3^{YFP-Cre}$ mice, we tested whether $T_{reg}$ cell-specific

ectopic expression of Id2 can alter anti-tumor immunity. In order to implement an experimental scheme in which $T_{reg}$-specific expression of Id2 commences simultaneously with tumor implantation, we employed an inducible model of temporal expression of Id2 in $T_{reg}$ cells by crossing $Id2^{EmGFP}Foxp3^{YFP-Cre}$ mice with $P_{CMV}$-TetR mice (resultant mouse called TetR-$Id2^{EmGFP}Foxp3^{YFP-Cre}$). In this model, under steady state condition, TetR binds to the TetO, repressing the expression of Id2-EmGFP protein. Addition of Dox releases this repression (Fig. 7a, b). Mouse melanoma B16.F10 model was induced and doxycycline or control PBS was administered according to the experimental scheme (Fig. 7c). Indeed, Dox-treated mice exhibited protective effect against tumor growth with significant decrease in tumor weight (Fig. 7d–f). Analysis of tumors revealed a correlation between reduced tumor volume and increased CD4+ and CD8+ T-cell infiltrates in Dox-treated animals (Supplementary Fig. 9a). More importantly, reduction in tumor progression

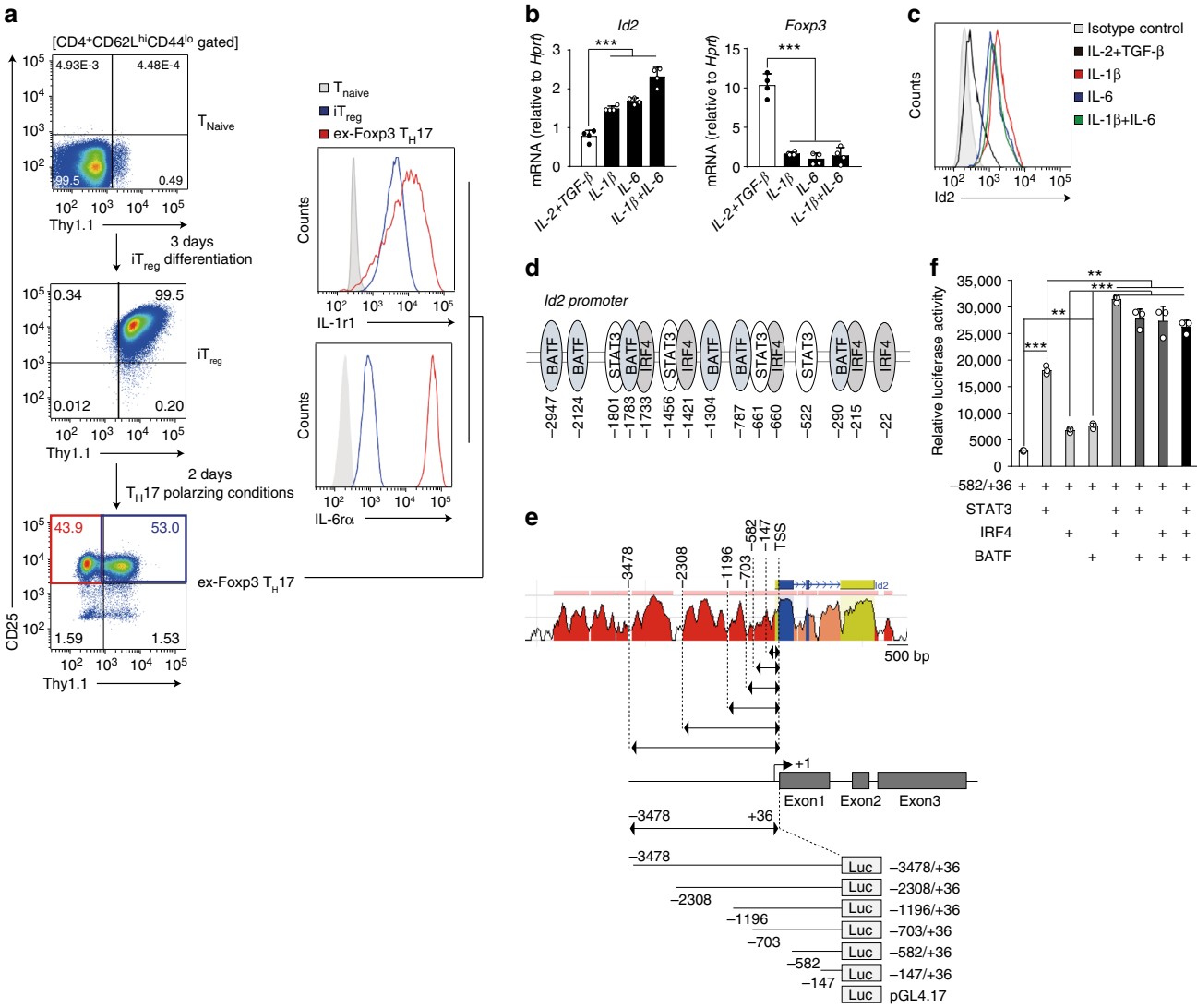

**Fig. 5** Id2 is induced by IL-1β and IL-6 mediated signaling and downstream transcription factors STAT3, IRF4 and BATF in T$_{reg}$ cells. **a** Strategy for sorting T$_{Naive}$, iT$_{reg}$, and ex-Foxp3 T$_H$17 cells from Foxp3$^{Thy1.1}$ reporter mice (left). Flow cytometry analysis of IL-1r1 and IL-6rα among the T$_{Naive}$, iT$_{reg}$, and ex-Foxp3 T$_H$17 cells (right). **b**, **c** FACS-sorted iT$_{reg}$ cells were re-stimulated with cytokines as indicated and analyzed for Id2 and Foxp3 mRNA expression by RT-qPCR (**b**) as well as for Id2 protein levels by flow cytometry (**c**). **d** Cartoon depicting highly putative STAT3, IRF4, and BATF binding motifs upstream of Id2 transcription start site (TSS). **e** Cartoon depicting Id2 promoter constructs used for luciferase reporter assay. **f** Id2 promoter-luciferase construct (−582/+36) were co-transfected with the combination of STAT3, IRF4 and BATF expressing vectors in HEK-293 T cells. Lysates were prepared 30 h after transfection, and luciferase activities were measured with the reporter activities normalized to renilla luciferase activity. *P < 0.05, **P < 0.005, ***P < 0.001 (Student's t-test). All data are representative three independent experiments with similar results (error bars, s.d.)

in Dox-treated animals was strongly associated with reduced CD4$^+$Foxp3$^+$ T$_{reg}$ cell infiltration within tumor tissue as well as in tumor-draining lymph nodes (dLN) (Fig. 7g). CD8$^+$ tumor infiltrating lymphocytes (TILs) as well as CD8$^+$ T cells isolated from dLN in Dox-treated group had significantly higher IFN-γ expression (Fig. 7h). Consistent with our previous results (Fig. 3i), Id2 overexpressing T$_{reg}$ cells showed higher IL-17A expression in TILs and dLN of Dox-treated tumor bearing mice (Supplementary Fig. 9b). Significant increase in the level of IFN-γ and IL-17A was also observed particularly in CD4$^+$Foxp3$^-$ TILs (Supplementary Fig. 9c).

Based on these results, we sought to examine whether ectopic Id2 expression may negatively affect T$_{reg}$ cell suppressive activity. Considering the observation that in tumor microenvironment of Dox-treated mice both CD4$^+$Foxp3$^-$ and CD8$^+$ populations displayed enhanced cytokine production, we performed in vitro

suppression assays with both CD4$^+$ and CD8$^+$ T naive populations in combination with sorted T$_{reg}$ cells from PBS-treated or Dox-treated animals (Supplementary Fig. 10a, b). Indeed T$_{reg}$ cells from Dox-treated mice displayed marginally reduced suppressive activity against proliferation of both CD4$^+$Foxp3$^-$ and CD8$^+$ T cells (Supplementary Fig. 10b). In agreement to these results, under identical conditions, the CD4$^+$Foxp3$^-$ cells were found to produce significantly increased level of IFN-γ and IL-17A. Similar trend in cytokine production was also observed in CD8$^+$ T cells, although the difference did not achieve significance (Supplementary Fig. 10c). Furthermore, as expected the Dox-treated T$_{reg}$ cells under such culture conditions displayed enhanced loss of Foxp3 (Supplementary Fig. 10d), suggesting that at least in part, the defect in suppressive activity observed in Id2 expressing T$_{reg}$ cells is due to instability in Foxp3 expression.

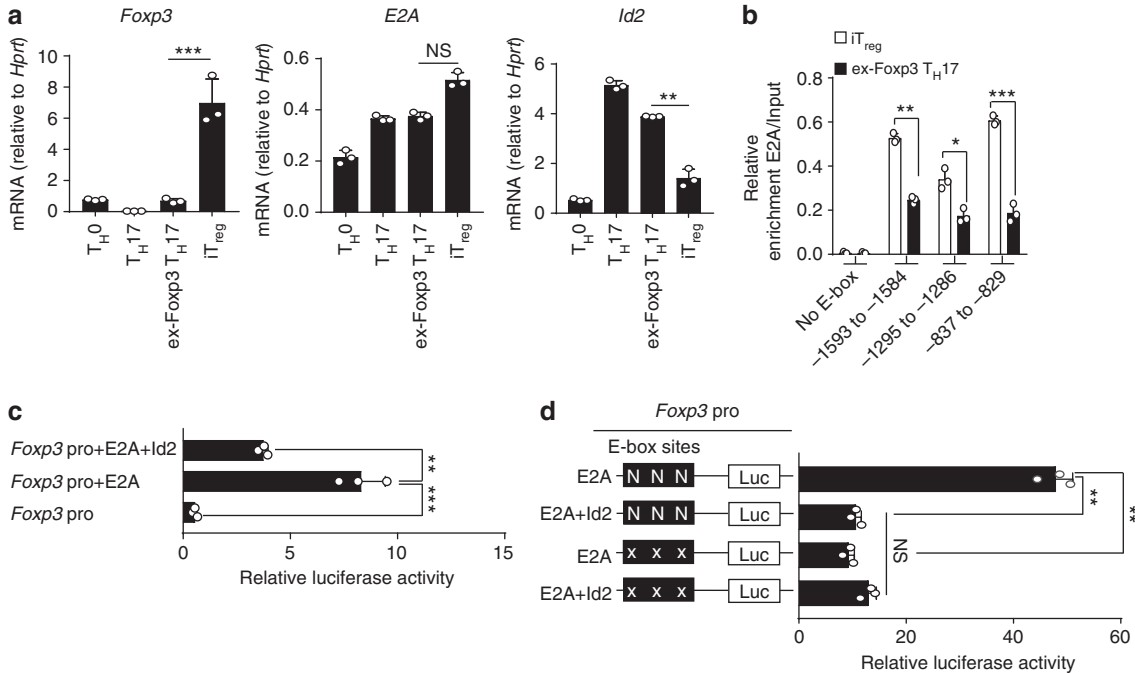

**Fig. 6** Id2 expression inhibits enrichment and transcriptional activity of E2A on *Foxp3* promoter. **a** RT-qPCR analysis of *Foxp3*, *E2A* and *Id2* mRNA in in vitro generated $T_H0$, $T_H17$, ex-Foxp3 $T_H17$, and $iT_{reg}$ cells. **b** ChIP-qPCR analysis for E2A occupancy at three putative E-box sites on Foxp3 promoter regions ($-1593$ to $-1584$, $-1295$ to $-1286$, and $-837$ to $-829$) and negative control ($-411$ to $-244$; No E-box site) in $iT_{reg}$ and ex-Foxp3 $T_H17$ cells. Enrichments are calculated relative to the input chromatin for corresponding sites. **c** Jurkat cells were transfected with either *Foxp3* promoter reporter construct alone or along with expression plasmids encoding E2A and Id2 as indicated, followed by luciferase assay. **d** Effect of mutagenesis on *Foxp3* promoter reporter activity. Jurkat cells were transiently co-transfected with the indicated plasmids for luciferase assay as described. N = Normal E box, X = Mutated E box. NS, not significant, $*P < 0.05$, $**P < 0.005$, $***P < 0.001$ (Student's $t$-test). All data are representative of two or three independent experiments (error bars, s.d.)

## Discussion

Since continuous expression of Foxp3 is required for efficient $T_{reg}$ function, it is evident that cell intrinsic and extrinsic mechanisms contributing to stability of Foxp3 are likely to influence multiple aspects of immune activation and tolerance in health and disease. While a number of *cis*-acting and *trans*-acting mechanisms have now been identified that positively contribute to transcription and maintenance of *Foxp3*[2], whether additional mechanisms exist in order to negatively influence Foxp3's expression in a context dependent manner, remains less clear. There is increasing evidence suggesting that $T_{reg}$ cells lose their lineage-stability and convert into diverse effector T cell phenotypes under certain inflammatory and lymphopenic conditions both in human and mice[11,12,14,15,26,43–45]. Under these conditions, $T_{reg}$ cells are readily converted to ex-Foxp3 $T_H17$ cells, impairing immune homeostasis and exacerbating certain immune disorders. In this study, by identifying Id2 to be critical for reprogramming $T_{reg}$ cells towards a $T_H17$ like phenotype in the context of inflammation, we have not only defined a hitherto unidentified mechanistic switch critical for inducing $T_{reg}$ plasticity, we have also identified a novel molecular target that can be potentially modulated in order to reinforce or undermine $T_{reg}$ stability in the context of autoimmunity and cancer.

In recent years, several studies demonstrate that TAZ (a coactivator of TEAD transcription)[46] and SGK (serum- and glucocorticoid-induced kinase 1)[47], as well as Socs1 (suppressor of cytokine signaling 1)[48] have critical roles in reciprocally regulating the differentiation of $T_{reg}$ and $T_H17$ cells. However, whether and how the expression of these factors is regulated under conditions promoting $T_{reg}$ plasticity is not fully understood. Here, we showed that Id2 is intrinsically reduced upon Foxp3 expression in $T_{reg}$ cells while highly expressed in $T_H17$

cells, implying opposite role of Id2 between $T_{reg}$ and $T_H17$ lineage compartments. In addition, Id2 is significantly enhanced during the conversion of $T_{reg}$ cells into ex-Foxp3 $T_H17$ cells under inflammatory conditions in vitro and in vivo, indicating that Id2 expression is critically associated with conversion of $T_{reg}$ into ex-Foxp3 $T_H17$ cells.

It is generally recognized that pro-inflammatory cytokines, such as IL-12, IL-1β, and IL-6 are the major contributors promoting inflammatory environments suitable for conversion of $T_{reg}$ cells into different types of ex-$T_{reg}$ cells under conditions of varied immune disorders such as ocular infection[43], helminth infection[49], or autoimmune diseases[13–15] (asthma, rheumatoid arthritis and multiple sclerosis). In particular, IL-1β and IL-6 mediated signaling cascade substantially reduce $T_{reg}$ stability and enhance the conversion of $T_{reg}$ cells to ex-Foxp3 $T_H17$ cells under physiological conditions[50]. We found that Id2 can be induced in $T_{reg}$ cells upon activation of IL-1β and IL-6 signaling, eventually leading to reduction in Foxp3 expression and promoting $T_H17$ related cytokine expression. Upon re-analysis of previously published high throughput ChIP-seq data, combined with luciferase assay based transcription analysis we implicated the transcription factors STAT3, BATF and IRF4, all of which are core members of the $T_H17$ transcription network, are involved in this process. However, what factor(s) contribute towards limiting the expression of Id2 in $T_{reg}$ cells is still unclear. Since green fluorescent protein (GFP) mediated functional knock-out of Foxp3, results in upregulation of Id2 in Foxp3$^{GFPKO}$ mice (ref[51] and our unpublished data), it seems likely that Foxp3 might act as a direct repressor of Id2 gene expression. However further experiments are required to confirm this possibility.

Upon its induction in pro-inflammatory conditions, Id2 was found to repress Foxp3 expression by sequestering the E-box

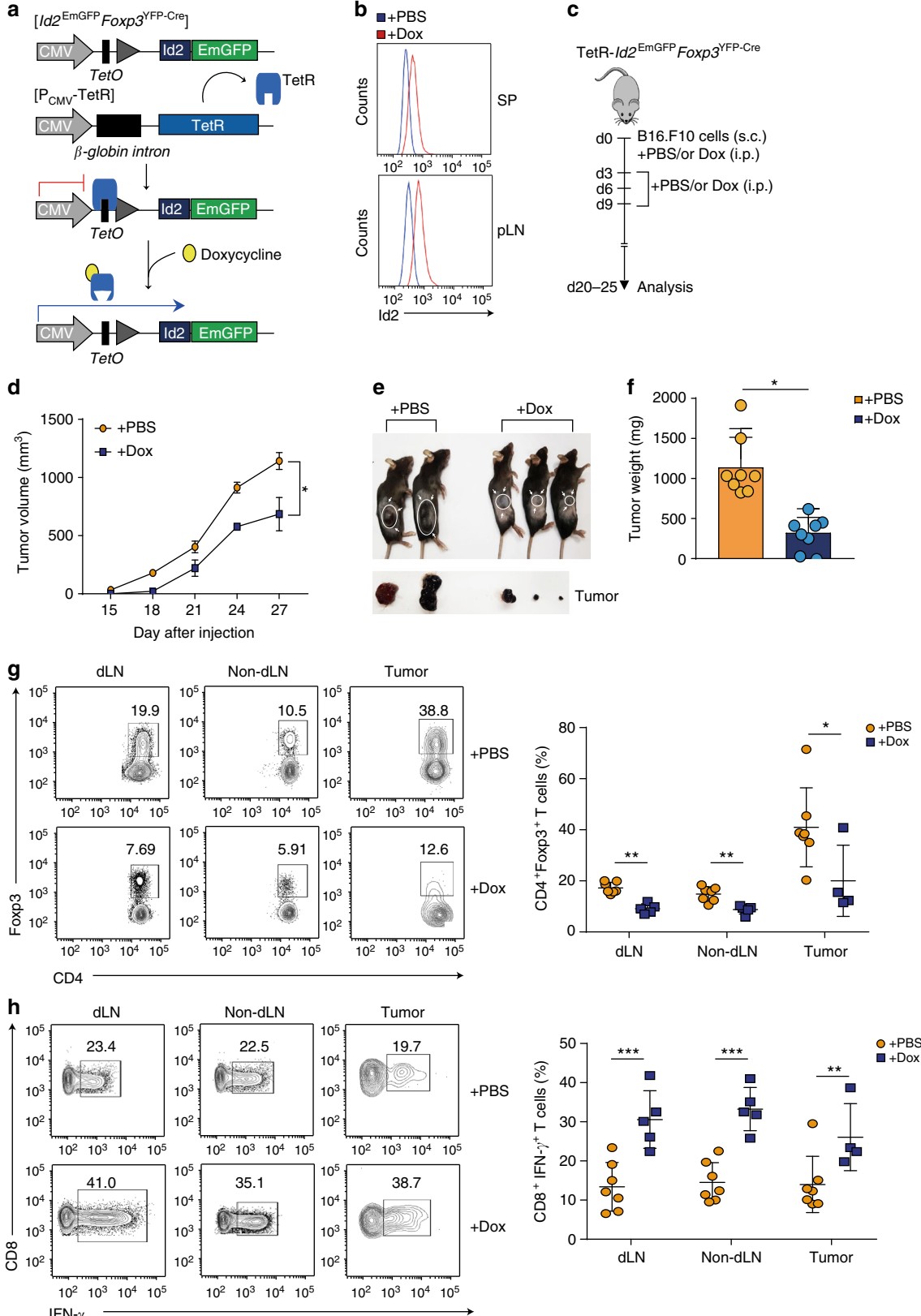

binding transcription factor E2A from binding to *Foxp3* promoter. E2A, a well-known member of the E-protein family, has been previously shown to promote transcription of *Foxp3* by physically associating with multiple regions of *Foxp3* promoter in T$_{reg}$ cells[32]. We found that *Foxp3* promoter contains three E-

protein binding sites in which E2A-mediated enhancement of *Foxp3* promoter activity could be significantly repressed upon concomitant expression of Id2. Of note, while well characterized conserved non-coding sequences (CNS1-3) within the *Foxp3* locus have been implicated in various aspects of Foxp3 expression

**Fig. 7** Id2 dependent $T_{reg}$ plasticity enhances antitumor immunity. **a** Schematic for doxycycline (Dox)-inducible Id2 overexpression in $T_{reg}$ cells in TetR-Id2$^{EmGFP}$Foxp3$^{YFP-Cre}$ mice. **b** Id2 expression was assessed by intracellular staining in SP and pLN from both PBS or Dox treated groups. **c** Experimental scheme of mouse melanoma model. Females, 8–10 week-old, TetR-Id2$^{EmGFP}$Foxp3$^{YFP-Cre}$ mice were injected subcutaneously with B16.F10 cells. Mice were treated intraperitoneally either with PBS or Dox from day 0–9, every 72 h. **d** Tumor progression, expressed as mean tumor volume (mm$^3$) in both treatment groups (PBS; $n = 8$, Dox; $n = 8$). **e** Representative tumor size in PBS and Dox treated groups on d 23. **f** Difference in tumor weight, as measured at end point of analysis (PBS; $n = 8$, Dox; $n = 8$). **g** Flow cytometry analysis of CD4$^+$Foxp3$^+$ T cells in tumor-draining lymph nodes (dLN), non-draining lymph nodes (non-dLN) as well as within tumor infiltrated lymphocytes. **h** Intracellular staining of cytokines in CD8$^+$ T cells isolated from dLN, non-dLN and tumor infiltrated lymphocytes, stimulated with phorbol myristate acetate (PMA) and ionomycin for 6 h. NS, not significant, *$P < 0.05$, **$P < 0.005$, ***$P < 0.001$ (Student's $t$-test). All data are representative of three independent experiments (error bars, s.d.)

as well as maintenance[3,5], they did not show any predicted E-protein binding sites, and were found to be dispensable for Id2 mediated repression of Foxp3 (data not shown). Since $T_{reg}$ cells make an immunosuppressive tumor microenvironment resulting in compromised anti-tumor immunity[33–35], substantial efforts are being made to reduce the number or function of $T_{reg}$ cells in tumor settings[38–42]. We found that ectopic expression of Id2 in $T_{reg}$ cells resulted in arrested tumor growth in a B16.F10 melanoma model, which was accompanied with loss of $T_{reg}$ cells and concomitant increase in IFN-γ producing cytotoxic T cells. Furthermore, consistent with our previous results, IL-17A was found to be significantly increased within the CD4$^+$Foxp3$^-$ tumor infiltrated lymphocytes, suggesting that these ex-Foxp3 $T_H$17 cells maintain their ability to produce pro-inflammatory cytokine. Collectively these results suggest that while increased $T_{reg}$ plasticity may result in detrimental cytokine production under inflammatory conditions, proper understanding of molecular events contributing to such instability can be harnessed in the context of cancer in order to boost anti-tumor immunity. We propose therefore that Id2 is one such molecule that can serve as an attractive target likely to be modulated to implement therapeutic modalities for $T_{reg}$ related disorders in the context or autoimmune disorders as well as cancer.

## Methods

**Mice**. Foxp3$^{YFP-Cre}$ [24] mice and Foxp3$^{Thy1.1}$ [23] mice are described elsewhere. P$_{CMV}$-lsl-Id2$^{EmGFP}$ and P$_{CMV}$-TetR transgenic mice were generated from Macrogen, Inc (Seoul, Korea). ROSA26$^{lsl-tdTomato}$ (R26T) mice were purchased from The Jackson Laboratory and were crossed with conditional Id2 transgenic mice to trace $T_{reg}$ cells stability and plasticity depend on $T_{reg}$-specific Id2 overexpression. All animals were maintained in a specific pathogen-free (SPF) conditions at POSTECH animal facility. All procedures were approved by the POSTECH Institutional Animal Care and Use Committee (IACUC).

**Microarray analysis**. Re-analysis was performed on previously published microarray gene-expression data[14] (Gene Expression Omnibus accession code GSE48428) to select putative target genes by comparing gene expression profile of ex-Foxp3 $T_H$17 (ex-$T_{reg}$) cells with $T_{reg}$ cells. We extracted target gene lists that were at least 2-folds up-regulated in ex-Foxp3 $T_H$17 cells originated from Foxp3$^+$ T cells ($P = 9.29e-92$ and $T_H$17 ($P = 1.84e-64$) compared to $T_{reg}$ cells and at least 2-folds down-regulated in $T_{reg}$ cells ($P = 4.97e-59$) compared to $T_H$0 cells. Similar Id2 expression patterns were confirmed from two different independent studies[15,22] (Gene Expression Omnibus accession code GSE80804 and GSE60059 respectively). Data were presented as heat map by using R-Studio.

**Flow cytometry and cell sorting**. Fixable Viability Dyes (eBioscience) were used at 1:1000 dilution to label dead cells. For surface staining, cells were stained with the following fluorescence-conjugated antibodies (eBioscience, Biolegend, Tonbo): anti-CD4 (RM4-5), anti-CD8 (53–6.7), anti-CD25 (PC61), anti-CD44 (IM7), anti-CD62L (MEL-14), anti-Thy1.1 (OX-7), IL-1R (JAMA-147), IL-6Rα (D7715A7) were used at 1:400 dilution. For intracellular staining, surface stained cells were fixed and permeabilized with a Foxp3 staining kit (eBioscience) according to manufacturer's instruction and were stained with the following antibodies: anti-Foxp3 (FJK-16s), anti-Rorγt (AFKJS-9), anti-Id2 (ab166708; Abcam), anti-IFN-γ (XMG1.2), anti-IL-17A (TC11-18H10.1), anti-IL-17F (eBio18F10), anti-IL-22 (1H8PWSR), anti-IL10 (JES5-16E3) were used at 1:200 dilution. For intracellular cytokine staining of cytokines, cells were stimulated by phorbol myristate acetate (PMA) and ionomycin for 6 h in the presence of Golgi-Plug (555029, BD) or Golgi-Stop (554724, BD). Data from the stained cells were collected with LSR Fortessa flow cytometer analyzer equipped with 5 lasers with DIVA software (BD

Biosciences) and were analyzed by FlowJo software (Treestar). Gating strategies for FACS sorting are described in Supplementary Fig. 11.

**In vitro CD4$^+$ T cell differentiation**. For CD4$^+$ T helper ($T_H$) cell differentiation, purified naive CD4$^+$CD25$^-$CD62L$^{hi}$CD44$^{lo}$ T cells from wild-type C57BL/6 (B6) mice or CD4$^+$CD25$^-$CD62L$^{hi}$CD44$^{lo}$Thy1.1$^-$ T cells from Foxp3$^{Thy1.1}$ mice (1 × 10$^6$/ml) were activated with plate-bound αCD3 (1 μg/ml; 1452C11; Bio Xcell) and αCD28 (2 μg/ml; clone 37.51; Bio Xcell) antibodies in the presence of appropriate cytokines. $T_H$0: anti-IL-4 (10 μg/ml), anti-IFN-γ (10 μg/ml; XMG 1.2; Bio Xcell) and 100U/ml of recombinant human IL-2(rhIL-2), $T_H$17: rIL-1β (20 ng/ml), rIL-6 (20 ng/ml), hTGF-β1 (2 ng/ml), anti-IL-4 (10 μg/ml) and anti-IFN-γ (10 μg/ml) or iT$_{reg}$: rhIL-2(100U/ml) and human TGF-β1 (hTGF-β1; 5 ng/ml) conditions in T cell media. 100 U/ml of rhIL-2 ($T_H$0 and iT$_{reg}$) and 30 U/ml of rhIL-2 and rIL-23 (10 ng/ml) ($T_H$17) were added on 3 days after detaching from αCD3/αCD28 antibodies, and the cells were expanded in complete T cell media additional for 2 days.

**RNA extraction and quantitative RT-qPCR**. Total RNA was extracted by TRI Reagent (Molecular Research Center, USA). For reverse transcription, cDNA was generated using 500 ng of total RNA, oligo(dT) primer (Promega) and Improm-II Reverse Transcriptase (Promega) in a total volume of 20 μl. One microliter of cDNA was amplified by SYBR Premix Ex Taq (Takara, Japan) using DNA Engine with Rotor-gene Q (Qiagen, Valencia, USA). Mouse hypoxanthine-guanine phosphoribosyl transferase (HPRT) was used for normalization. All primers used in this study are summarized in Supplementary Table 1.

**Induction and assessment of EAE**. Mice were immunized subcutaneously with 200 μg/ml of MOG 33−55 peptide (AnyGen, Korea) with CFA containing 4 mg/ml heat-killed M. tuberculosis. At 0 days and 2 days, 2 μg/ml of pertussis toxin suspended in PBS was injected to immunized mice by intraperitoneal injection. Clinical scores were monitored every other day (0: Normal behavior, 1: Tail paralysis, 2: Hind legs paralysis, 3: Front legs paralysis, 4: Full paralysis, 5: Death).

**Histology**. Tissues were fixed in 4% paraformaldehyde. Fixed tissues were embedded in paraffin and sectioned at 3μm thickness, followed by hematoxylin (Sigma-Aldrich, MO, USA) and eosin (Sigma-Aldrich, MO, USA) (H&E) staining. The images were captured by Leica DFC425 C microscopy.

**Plasmids and retroviral transduction**. Retroviruses were produced by transfecting 6 μg of the retroviral expression vector together with 4 μg of the retroviral packaging vector (pCL-Eco) into Plat-E packaging cells in culture media (DMEM+ 10% FBS). Forty-eight hours after transfection, high-titer viral supernatant was collected. Each differentiating CD4$^+$ T cells was transduced with control (MigR1-GFP) or Id2 expressing retrovirus (MigR1-Id2-GFP) at 24 h after in vitro stimulation. The transduction was performed in a 12-well plate by spinning for 90 min at 2500 rpm, room temperature in the presence of 8 μg/ml of polybrene. The transduced cells were analyzed 3 days after the start of culture.

**Western blot analysis**. Cells (1–5 × 10$^6$) from in vitro generated $T_H$17, iT$_{reg}$ and ex-Foxp3 $T_H$17 cells from $T_{Naive}$ (CD4$^+$CD62L$^{hi}$CD44$^{lo}$) cells were lysed using RIPA lysis buffer and separated by SDS-PAGE. Proteins were detected using the following antibodies: Primary, anti-Mouse Foxp3 (FJK-16s, eBioscience), anti-Mouse Roγrt (Rorg2, Biolegend), anti-Mouse Id2 (166708, Abcam), anti-Mouse beta Actin (8226, Abcam); Secondary, anti-Rabbit IgG HRP (AbC-5003, AbClon), anti-Armenian Hamster IgG HRP (2443, Santa Cruz), anti-Mouse IgG HRP (AbC5001, Abclon). Images were captured by using a ImageQuant$^{TM}$ LAS4000 (GE Healthcare). Complete scanned gels for western blots are presented in Supplementary Fig. 12.

**Plasmids and luciferase reporter assays**. HEK-293 T cells or Jurkat cells were transfected using FuGENE® HD (Promega) or Nucleofector$^{TM}$ (Lonza) according to the manufacturer's protocol and plated in 12 well plate. 0.5 μg of the Id2 promoter-luciferase constructs or pGL4.17 empty vector was co-transfected with

STAT3 (MR227265; Origene), IRF4 (MR226642; Origene) and BATF (MR222114; Origene) expression plasmids (HEK-293 T) and 1 µg of the *Foxp3* promoter-luciferase construct was co-transfected with E2A (MG209745; Origene) and Id2 (MR200792; Origene) expression plasmids (Jurkat). Lysates were prepared 30 h after transfection, and luciferase activities were measured with the Dual-Luciferase Reporter Assay System (Promega). The reporter activities were normalized to renilla luciferase activity. Mutations of E2A binding sites were done by using the GeneArt™ Site-Directed Mutagenesis PLUS System kit (Invitrogen).

**Chromatin immunoprecipitation (ChIP)**. CD4$^+$Foxp3$^+$ iT$_{reg}$ cells (5–7 × 10$^6$) or CD4$^+$Foxp3$^−$ ex-Foxp3 T$_H$17 cells originated from iT$_{reg}$ cells (5–7 × 10$^6$) were used. An equal amount of processed chromatin was used as an input control or was incubated with 4 µg antibody to purified mouse anti-human E47(G127-32; BD Biosciences). Immunoprecipitated DNA and total input DNA were analyzed with SYBR Premix Ex Taq (Takara, Japan) using DNA Engine with Rotor-gene Q (Qiagen, Valencia, USA). PCR primers for the detection of each *Foxp3* promoters were as follows: *Foxp3* amplicon 1 (−411 to −244; No E-box binding sites), 5′-GGATGCCTTTGTGATTTGAC-3′ (forward) and 5′-TTTGCCCTTTACAAGTC ATCTG-3′ (reverse); *Foxp3* amplicon 2 (−1593 to −1584; E-box BS1), 5′-GATACCTGGAACTCTTAGCTC-3′ (forward) and 5′-GTCATAGAAGTTCT AGGACTTGG-3′ (reverse); and *Foxp3* amplicon 3 (−1295 to −1286; E-box BS2), 5′-AACAATACAGCCATGATGAGATGGA-3′ (forward) and 5′-GCAAAGGTTT AGGATTCTAAACAGC-3′ (reverse). *Foxp3* amplicon 4 (−837 to −829; E-box BS3), 5′-TTGCCCTTCTTGGTGATGCT-3′ (forward) and 5′- CATGTTTGTGCT GAGTGCCC-3′ (reverse).

**Mouse melanoma model**. To establish in vivo mouse tumor model, 0.2 × 10$^6$ B16. F10 cells, obtained from American Type Culture Collection (ATCC, USA) were subcutaneously injected in the flank region, on day 0. Inducible Id2 transgenic mice (TetR-*Id2*$^{EmGFP}$*Foxp3*$^{YFP−}$Cre) were treated with intra-peritoneal injections of either PBS or Doxycycline (50 µg/g mice in 200 µl) on day 0, 3, 6, and 9 to induce Id2 expression in T$_{reg}$ cells. Tumor growth was monitored on alternate days, from day 15 through day 27 post tumor cells injection. Vernier calipers were used for measuring length and width of the tumors and tumor volume was calculated using formula Volume (mm$^3$) = (length × width × width)/2. Euthanasia was performed at the end of experiment or if tumor volume exceeded 2000 mm$^3$ or the tumor growth obstructed the feeding of mice.

**In vitro suppression assay**. CD4$^+$Foxp3$^+$ T$_{reg}$ cells sorted from PBS and Dox treated mice were washed three times with PBS and incubated with responder cells (CD45.1$^+$CD62L$^{hi}$CD44$^{lo}$Foxp3$^−$CD4$^+$ or CD8$^+$) that were pulsed in 1 ml of PBS with 1uL of a 5 µM CTV stock for 10 min at 37 °C. CTV labeled cells were washed in PBS twice and immediately used. 1 × 10$^5$ APC depleted of CD4 and CD8 splenocytes by negative selection (Miltenyi Biotech) were mixed with 5 × 10$^4$ CTV-pulsed Foxp3$^−$CD4$^+$ or CD8$^+$ responder cells. T$_{reg}$ and responder cells ratio used as indicated within the figure along with 0.4µg/ml of anti-CD3 in a round bottom 96-well plate. Cultures were incubated for 4 days and then analyzed by flow cytometry to determine CTV dilution as well as intracellular staining for cytokines after stimulation with phorbol myristate acetate (PMA) and ionomycin for 6 h in the presence of Golgi plug.

**Statistical analysis**. Statistical tests were performed using Prism 7.03 software (GraphPad). Significant differences were measured by unpaired, two-tailed Student's *t*-test with a 95% confidence interval. Statistical significance was considered as $P < 0.05$ (*<0.05, **<0.005, ***<0.001).

**Reporting Summary**. Further information on research design is available in the Nature Research Reporting Summary linked to this article.

## Data availability

Previously published microarray data supporting the findings of this study have been deposited in the Gene Expression Omnibus and are publicly available under the GEO accession number GSE48428, GSE80804 and GSE60059. All data that support the findings of this study are available from the corresponding authors upon request.

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

## Acknowledgements

We thank to H. J. Jung and M. O. Lee for technical assistance for cell sorting, and Jonathan Sprent and Yun Kyung Lee for helpful discussions and comments on this study. This research was supported by Institute for Basic Science (Project Code: IBS-R005-D1), and by Global Ph.D. Fellowship Program through the National Research Foundation of Korea (NRF) funded by the Ministry of Education (Project number: 2015H1A2A1030032).

## Author contributions

D.R., S.-H.I. and S.-M.H. designed the experiments and S.-M.H. performed most of the experiments and analyzed the data. G.S. assisted with in vivo tumor models and wrote the corresponding section of the manuscript. R.V. performed the in vitro suppression assays. S.B. performed the blinded clinical scoring on EAE disease and assisted in vitro and in vivo experiments. D.R. and S.-H.I. co-supervised the study. D.R., S.-H.I. and S.-M.H. wrote the manuscript.

## Additional information

**Competing interests:** The authors declare no competing financial or non-financial interests.

