## [Peer Review file · Nature Communications]

Reviewers' comments:

Reviewer #1 (T cell immunity and tolerance)(Remarks to the Author):

In this paper, the authors have investigated a role of Id2 in the stability and plasticity of Tregs. They showed with multiple experimental approaches that Id2 expression in Tregs resulted in downregulation of Foxp3 expression and consequentially increase /conversion to IL-17 production. Importantly, by generating Treg-specific Id2 conditional knockout mice, the authors have demonstrated the importance of Id2 in the destabilization of Treg phenotype and function in EAE and experimental tumor models. The paper contains important and interesting findings. I do not have major concerns. A few relatively minor questions need to be addressed to further increase the impact of the paper.

1, Fig. 1, in the experimental scheme for iTreg conversion to ex-Foxp3 Th17 cells, the authors indicate to use Th17 polarization conditions. Was exogenous TGF- β 1 included in the cocktail of the cytokines? If yes, what is the rationale to include TGF- β 1? If the authors believe TGF- β is important in this Th17 polarization condition to convert iTreg to ex-foxp3 Th17 cells, a control culture with IL-1 β +IL-6+aIL-4+aIFN- γ (without TGF- β 1) should be included. Please clarify.

2, also in Fig. 1e, it seems that ROR γ t cannot be induced during this culture, despite of Id2 induction. Does this mean that the acquisition of IL-17 from iTregs does not need ROR γ t?

3, Fig.7, do CD4+ IFN- γ + (Th1)-like cells also increase in Id2-overexpressing mice in this tumor model? Is the increase in CD8+IFN- γ + T cells due to the lack of Treg suppression because of Id2? some kinds of functional studies are needed in these Id2+ Tregs.

Reviewer #2 (Th subsets, plasticity)(Remarks to the Author):

In the manuscript by Sung-Min Hwang et al "Inflammation-induced Id2 promotes plasticity in regulatory T cells" the authors demonstrate that Id2 induced by the inflammatory cytokines IL-1 β and IL-6 is able to interact with E2A transcription factor inducing its sequestration and consequent inhibition of Foxp3 expression. Upon upregulation of Id2, Treg cells acquire the ability to produce both IL-17 and IFN- γ and associate with increased pathogenicity in a model of EAE. The manuscript is of interest but some major points need to be fully addressed.

1) The experiments performed in vitro need some clarification.

a) Fig.1: It is important to show the frequencies of FOXP3 and IL-17a producing cells under the different culture conditions. Looking at the frequency of FOXP3 expressing cells shown in Fig 2c under iTreg culturing condition, only 40% of the cells stain positive for this marker. This make difficult to define which are the cells that once moved into Th17 culturing conditions polarize into "exTreg" (FOXP3+ or FOXP3- cells?). Why the author did not take advantage of naïve CD4 T cells isolated from FOXP3-YFP Δ mice? They can in this case sort YFP+ cells obtained by in vitro polarization into Treg and further culture these cells into Th17 polarizing condition

b) Supplementary Fig 2: During the Th17 differentiation of naïve CD4 T cells the author did not show any data performed at single cell level demonstrating the effectiveness of the polarization cocktail. In particular looking at fig 2 b of supplementary data, rorgt expression, as assessed by qPCR, reaches levels comparable to other master transcription factor specific of other T cell subsets. The authors need to perform a flow cytometric evaluation of intracellular cytokines. Moreover, also Treg differentiation

2) The observation made in point 1 raises one concern: since the in vitro polarizing conditions did not

induce an omogeneous differentiation of all the cells into that particular cell subset, the authors cannot make any conclusion on results obtained upon transduction of Id2 made on day 1. I would suggest to first transduce the naïve CD4 T cells and then start the polarizing process.

3) Are other cytokines in addition to IL-17 (IFN-g, TNF-a.....) affected by ID2 overexpression during iTreg polarization?

4) In the manuscript by Takahisa Miyao et "Plasticity of Foxp3+ T Cells Reflects Promiscuous Foxp3 Expression in Conventional T Cells but Not Reprogramming of Regulatory T Cells", the authors showed that transient FoxP3 expression can be induced upon activation of murine CD4 T cells (this was already largely demonstrated in human). I'm wondering if the Tomato+ cell subset really identify ex Treg (YFP-) and Treg (YFP+), or if part of the Tomato+YFP- cells represent previously activated conventional T cells. The authors, in order to fully support their conclusion, need to clarify this point

Rebuttal Letter:

A point-by-point response to reviewers' comments

First of all, we are thankful to all the reviewers for the encouraging and positive comments to improve our manuscript and give us the opportunity to submit revised version of our manuscript NCOMMS-18-17418-T entitled "Inflammation-induced Id2 promotes plasticity in regulatory T cells". We have taken into account all the points raised by reviewers and performed further experiments to clarify those. According to reviewers' comments, we agree with almost all their comments and we have revised our manuscript accordingly. Please, find below the reviewers' comments repeated in *italics* and **our responses** inserted after each comment.

Reviewer 1

※ General comment:

In this paper, the authors have investigated a role of Id2 in the stability and plasticity of Tregs. They showed with multiple experimental approaches that Id2 expression in Tregs resulted in downregulation of Foxp3 expression and consequentially increase /conversion to IL-17 production. Importantly, by generating Treg-specific Id2 conditional knockout mice, the authors have demonstrated the importance of Id2 in the destabilization of Treg phenotype and function in EAE and experimental tumor models. The paper contains important and interesting findings. I do not have major concerns. A few relatively minor questions need to be addressed to further increase the impact of the paper.

► Specific comment #1:

Fig. 1, in the experimental scheme for iTreg conversion to ex-Foxp3 Th17 cells, the authors indicate to use Th17 polarization conditions. Was exogenous TGF- β 1 included in the cocktail of the cytokines? If yes, what is the rationale to include TGF- β 1? If the authors believe TGF- β 1 is important in this Th17 polarization condition to convert iTreg to ex-foxp3 Th17 cells, a control culture with IL-1 β +IL-6+ α IL-4+ α IFN- γ (without TGF- β 1) should be included. Please clarify.

Response #1:

We apologize for not being clear. For iT_{reg} conversion to ex-Foxp3 T_H17 cells, we have washed the cultured iT_{reg} cells and treated with mixture of T_H17 polarizing cytokines (IL-1 β +IL-6+TGF- β 1+ α IL-4+ α IFN- γ). Since it has been reported that TGF- β 1 and IL-6 are required for polarization of T_H17 cells¹⁻³, we have included exogenous TGF- β 1 in the cocktail of cytokines to convert iT_{reg} to ex-Foxp3 T_H17 cells. By following reviewer's comment, we have also repeated this experiment and included a control experiment with IL-1 β +IL-6+ α IL-4+ α IFN- γ (without TGF- β 1), followed by intracellular staining for IL-17A and IL-10 as well as

relevant transcription factors (new Supplementary Fig. 2c, and corresponding text highlighted in lines 136-140). While TGF- β 1 was dispensable for Id2 upregulation and Foxp3 downregulation, we indeed observed less IL-17A production from ex-Foxp3 T_H17 cells under conditions devoid of TGF- β 1, suggesting that similar to conditions promoting T_H17 from T naïve, presence of TGF- β 1 is required for optimum conversion to ex-Foxp3 T_H17 cells from iT_{reg} cells as well.

► **Specific comment #2:**

also in Fig. 1e, it seems that ROR γ t cannot be induced during this culture, despite of Id2 induction. Does this mean that the acquisition of IL-17 from iTregs does not need ROR γ t?

Response #2:

We thank the reviewer for this insightful comment. The expression kinetics and subsequent transcriptional program exerted by Ror γ t appears to commence within a definite window during T naïve to T_H17 differentiation process⁴ (Fig. 1c). It indeed appears that in contrast to T naïve to T_H17 differentiation, Ror γ t expression to a great extent is dispensable for iT_{reg} to ex-Treg T_H17 conversion. It may be possible that unlike T naïve cells, previous exposure to TGF β for the precursor iT_{reg} cells in this case was enough to prepare the transcriptional landscape of these cells to bypass the requirement of Ror γ t for subsequent conversion to T_H17 cells. However, provided the miniscule increase in Ror γ t observed only in TGF β containing condition (Fig. 1e and Supplementary Fig. 1b lower panel), it might be possible that the TGF β -dependent advantage in T_H17 related cytokine production in this case (Supplementary Fig. 2c) is to some extent dependent on Ror γ t.

► **Specific comment #3:**

Fig.7, do CD4+ IFN-g+ (Th1)-like cells also increase in Id2-overexpressing mice in this tumor model? Is the increase in CD8+IFN-g+ T cells due to the lack of Treg suppression because of Id2? some kinds of functional studies are needed in these Id2+ Tregs.

Response #3:

We really appreciate for the reviewer's comments and apologize for not being clear earlier. In our initial submission (previous Supplementary Fig 6c and new Supplementary Fig. 9c), we have included the data demonstrating enhance T_H1 and T_H17 cytokine production from CD4⁺Foxp3⁻ compartment of Doxycycline treated TetR-Id2^{EmGFP} mice particularly in tumor

and tumor draining lymph nodes (dLN). The corresponding text stating this observation is highlighted in lines (343-344).

By following reviewer's suggestion, we have performed *in vitro* suppression assay with the sorted T_{reg} cells from both PBS- and Dox-treated mice using both CD4⁺Foxp3⁻ and CD8⁺ T cells as responder populations in separate assays. T_{reg} cells from Dox-treated mice displayed marginally reduced suppressive activity against naïve CD4⁺ and CD8⁺ T cells, which displayed enhanced proliferation as well as increased cytokine production. At least a part of this reduced suppressive capacity in Doxycycline treated T_{reg} cells appear to be due to enhanced loss of Foxp3. This experiment is demonstrated in Supplementary Fig. 10 and corresponding text highlighted in lines 346-359.

Reviewer 2

※ General comment:

In the manuscript by Sung-Min Hwang et al “Inflammation-induced Id2 promotes plasticity in regulatory T cells” the authors demonstrate that Id2 induced by the inflammatory cytokines IL-1beta and IL-6 is able to interact with E2A transcription factor inducing its sequestration and consequent inhibition of Foxp3 expression. Upon upregulation of Id2, Treg cells acquire the ability to produce both IL-17 and IFN-gamma and associate with increased pathogenicity in a model of EAE. The manuscript is of interest but some major points need to be fully addressed.

▶ Specific comment #1:

The experiments performed in vitro need some clarification.

a. Fig.1: It is important to show the frequencies of FOXP3 and IL-17a producing cells under the different culture conditions. Looking at the frequency of FOXP3 expressing cells shown in Fig 2c under iTreg culturing condition, only 40% of the cells stain positive for this marker. This make difficult to define which are the cells that once moved into Th17 culturing conditions polarize into “exTreg” (FOXP3+ or FOXP3- cells?). Why the author did not take advantage of naïve CD4 T cells isolated from FOXP3-YFPCre mice? They can in this case sort YFP+ cells obtained by in vitro polarization into Treg and further culture these cells into Th17 polarizing condition

Response #1a:

We appreciate the reviewer's comment and suggestion and we agree that performing this experiment using sorted T naïve cells from a mouse strain in which T_{reg} cells are marked, would lead to more robust interpretations. We repeated this experiment by using Foxp3^{Thy1.1}

reporter mice in which the Thy1.1 allele is knocked-in into *Foxp3* locus⁵. Naïve T cells were isolated from *Foxp3*^{Thy1.1} reporter mice; T_{reg} cells were induced for 3 days and Thy1.1⁺ iT_{reg} cells were sorted for further conversion into the ex-*Foxp3* T_H17 cells under T_H17 polarizing conditions, followed by FACS analyses for relevant transcription factors and cytokines. This experiment, which showed identical results as Fig. 1, is included in Supplementary Fig. 2 (highlighted text 129-140).

b. Supplementary Fig 2: During the Th17 differentiation of naïve CD4 T cells the author did not show any data performed at single cell level demonstrating the effectiveness of the polarization cocktail. In particular, looking at fig 2 b of supplementary data, rortg expression, as assessed by qPCR, reaches levels comparable to other master transcription factor specific of other T cell subsets. The authors need to perform a flow cytometric evaluation of intracellular cytokines. Moreover, also Treg differentiation

Response #1b:

We understand reviewer's concern. In response, we would first like to humbly point out that according to a previous report; the expression of *Rortg* appears to be regulated within a tight window during T_H17 differentiation process, which after robust induction early on, is greatly reduced over time⁴. In our initial experiment, we had measured the mRNA level of *Rortg* at day 3 (72hrs), at which point presumably its expression was reduced to the level comparable to the other transcription factors. To clarify this issue, we have additionally performed RT-PCR analysis at an early stage (12 hrs) of T_H17-induction as well. Indeed we observed increased *Rortg* expression in both vector transduced and *Id2* overexpression groups at this point. In contrast, while *Rortg* expression was reduced at 72 hrs in vector transduced cells, the provision of *Id2* appeared to delay the process (Supplementary Fig. 3b). In addition, by following reviewer's suggestions, we have performed flow cytometric analyses of intracellular cytokines for T_H17 cells, alongside with T_{reg} differentiation condition shown previously in Fig. 2e (new Supplementary Fig. 3e and highlighted lines 149-155, 159-160). Interestingly, while IL-17F and IL-22 expression is seen to be increased significantly at this point in terms of protein level, IL-17A expression, albeit increase in mRNA, was comparable between the two groups, presumably reflecting a delay in translation (Supplementary Fig. 3e).

► Specific comment #2:

The observation made in point 1 raises one concern: since the in vitro polarizing conditions did not induce an homogeneous differentiation of all the cells into that particular cell subset,

the authors cannot make any conclusion on results obtained upon transduction of Id2 made on day 1. I would suggest to first transduce the naïve CD4 T cells and then start the polarizing process.

Response #2:

We appreciate the comments and insights and agree with the reviewer's concern. Following reviewer's suggestion, we repeated this experiment using a modified protocol whereby T naïve cells were activated in the absence of differentiation conditions for one day prior to retroviral transduction. Differentiation conditions were introduced after spinfection. While under this experimental condition we repeatedly obtained much lower extent of differentiation particularly for the transduced GFP⁺ cells, the overall results remained the same. We have included this experiment in Supplementary Fig. 4, and corresponding text highlighted in lines 161-170.

► **Specific comment #3:**

Are other cytokines in addition to IL-17 (IFN-g, TNF-a.....) affected by ID2 overexpression during iTreg polarization?

Response #3:

In order to address reviewer's query we performed RT-PCR analyses to detect a panel of cytokines in cells harboring empty vector or Id2 overexpressing retrovirus. We found that along with T_H17 related cytokines, IFN- γ was increased in iT_{reg} cells harboring Id2 RV. T_H2 related cytokines IL-4, IL-5 and IL-13 as well as IL-10 and TGF- β and TNF- α remained unchanged (new Fig. 2d). These results are described in the text (highlighted lines 159-160). Additionally, we have performed same experiment during T_H17 polarization condition, in which case only the T_H17 related cytokines were the only ones increased and IFN- γ was not changed. These results are shown in the new Supplementary Fig. 3d.

► **Specific comment #4:**

In the manuscript by Takahisa Miyao et "Plasticity of Foxp3+ T Cells Reflects Promiscuous Foxp3 Expression in Conventional T Cells but Not Reprogramming of Regulatory T Cells", the authors showed that transient FoxP3 expression can be induced upon activation of murine CD4 T cells (this was already largely demonstrated in human). I'm wondering if the Tomato+ cell subset really identify ex Treg (YFP-) and Treg (YFP+), or if part of the

Tomato+YFP- cells represent previously activated conventional T cells. The authors, in order to fully support their conclusion, need to clarify this point.

Response #4:

We thank the reviewer for this insightful comment. We agree that the source of increased tdTomato⁺YFP⁻ cells upon EAE induction in *Id2^{EmGFP}R26TFoxp3^{YFP-Cre}* mice could be due to increased stability of promiscuous Foxp3 expressing T naïve cells. To clarify this point, we have performed the following experiments. Sorted naïve CD4⁺ T (CD4⁺tdTomato⁻YFP⁻ CD62L^{hi}CD44^{lo}) cells from *R26TFoxp3^{YFP-Cre}* and *Id2^{EmGFP}R26TFoxp3^{YFP-Cre}* were activated and cultured in the presence of anti-TGF-β. The Foxp3 expressing T cells under this condition have been reported as transient Foxp3 expressing cells⁶. However, the extent of promiscuous Foxp3 expressing T cells generated was found to be comparable between the two groups. On the contrary, a side by side assay in the presence of TGFβ, as expected, gave rise to increased frequency of iT_{reg} cells specifically in *R26TFoxp3^{YFP-Cre}* group. Furthermore, when CD4⁺FITC⁺tdTomato⁺ T_{reg} cells were sorted from the two groups and cultured *in vitro* for 4 days in the presence of IL-2, we observed enhanced loss of Foxp3 from *Id2^{EmGFP}R26TFoxp3^{YFP-Cre}* derived T_{reg} cells. Taken together these experiments strengthened the notion that the source of increased tdTomato⁺YFP⁻ cells in *Id2^{EmGFP}R26TFoxp3^{YFP-Cre}* are indeed T_{reg} cells that have downregulated Foxp3 expression. The experiments are included in Supplementary Fig. 10, and corresponding text highlighted in lines 220-237.

References

1. Bettelli, Estelle, et al. "Reciprocal developmental pathways for the generation of pathogenic effector TH17 and regulatory T cells." *Nature* **441**.7090 (2006): 235.
2. Mangan, Paul R., et al. "Transforming growth factor-β induces development of the TH17 lineage." *Nature* **441**.7090 (2006): 231.
3. Veldhoen, Marc, et al. "TGFβ in the context of an inflammatory cytokine milieu supports de novo differentiation of IL-17-producing T cells." *Immunity* **24**.2 (2006): 179-189.
4. Ivanov, Ivaylo., et al. "The orphan nuclear receptor RORγt directs the differentiation program of proinflammatory IL-17+ T helper cells." *Cell* **126**.6 (2006):1121-1133.

5. Liston, A. *et al.* Differentiation of regulatory Foxp3+ T cells in the thymic cortex. *Proceedings of the National Academy of Sciences* **105**, 11903-11908 (2008).
6. Miyao, T. *et al.* Plasticity of Foxp3(+) T cells reflects promiscuous Foxp3 expression in conventional T cells but not reprogramming of regulatory T cells. *Immunity* **36**, 262-275 (2012).

REVIEWERS' COMMENTS:

Reviewer #1 (Remarks to the Author):

The authors have adequately addressed my questions and comments.

Reviewer #2 (Remarks to the Author):

THE AUTHORS ADEQUATELY ADDRESSED ALL THE POINTS